# Lower Plasmatic Levels of Saturated Fatty Acids and a Characteristic Fatty Acid Composition in the Ovary Could Contribute to the High-Fertility Phenotype in Dummerstorf Superfertile Mice

**DOI:** 10.3390/ijms231810245

**Published:** 2022-09-06

**Authors:** Michela Calanni-Pileri, Joachim M. Weitzel, Dirk Dannenberger, Martina Langhammer, Marten Michaelis

**Affiliations:** 1Institute of Reproductive Biology, Research Institute for Farm Animal Biology (FBN), FBN Dummerstorf, Wilhelm-Stahl-Allee 2, 18196 Dummerstorf, Germany; 2Institute of Muscle Biology and Growth, Research Institute for Farm Animal Biology (FBN), 18196 Dummerstorf, Germany; 3Institute of Genetics and Biometry, Service Group Lab Animal Facility, Research Institute for Farm Animal Biology (FBN), 18196 Dummerstorf, Germany

**Keywords:** body fat content and fertility, high-fertility phenotype, superfertile mice, fatty acid profile, n-3 PUFA, n-6 PUFA, biodiversity

## Abstract

In recent decades, fertility traits in humans as well as in farm animals have decreased worldwide. As such, it is imperative to know more about the genetics and physiology of increased or high fertility. However, most of the current animal models with reproductive phenotypes describe lower fertility or even infertility (around 99%). The “Dummerstorf high-fertility lines” (FL1 and FL2) are two unique mouse lines selected for higher reproductive performances, more specifically for higher number of pups per litter. We recently described how those superfertile mice managed to increase their reproductive phenotype by doubling the ovulation rate and consequently the litter size compared to the unselected mice of the same founder population. FLs show an unusual estrous cycle length and atypical levels of hormones that link reproduction and metabolism, such as insulin in FL1 and leptin in FL2. Moreover, we described that their higher ovulation rate is mostly due to a higher quality of their oocytes rather than their sheer quantity, as they are characterized by a higher quantity of high-quality oocytes in antral follicles, but the quantity of follicles per ovary is not dissimilar compared to the control. In the present study, we aimed to analyze the lipid composition of the fertility lines from plasma to the gonads, as they can connect the higher reproductive performances with their metabolic atypicalities. As such, we analyzed the fat content of FLs and fatty acid composition in plasma, liver, fat, oocytes of different quality, and granulosa cells. We demonstrated that those mice show higher body weight and increased body fat content, but at the same time, they manage to decrease the lipid content in the ovarian fat compared to the abdominal fat, which could contribute to explaining their ovarian quality. In addition, we illustrate the differences in fatty acid composition in those tissues, especially a lower level of saturated fatty acids in plasma and a different lipid microenvironment of the ovary. Our ongoing and future research may be informative for farm animal biology as well as human reproductive medicine, mostly with cases that present characteristics of lower fertility that could be reversed following the way-of-managing of Dummerstorf high-fertility lines.

## 1. Introduction

### 1.1. Dummerstorf High-Fertility Mouse Lines

In the 1970s, the Institute for Farm Animal Biology (FBN—Dummerstorf, Germany) established a founder mouse line by cross breeding four inbred and four outbred lines, called FZTDU (Forschungszentrum für Tierproduktion Dummerstorf, Research Center for Animal Production Dummerstorf). This population represents the founder line that served as a starting point of a unique breeding program with five mouse lines (Dummerstorf selection lines) selected for different traits, plus the unselected control line. Among those, high-fertility lines, fertility line 1 (FL1) and fertility line 2 (FL2), were selected for a higher number of pups per litter [1]. After approximately 200 generations, both lines doubled the number of ovulated oocytes per cycle and, consequently, the litter size compared to the unselected control animals [2,3,4]. Of note, FLs have different endocrine as well as physiological parameters, such as initiation of puberty or life expectancy, as well as behavioral parameters, such as endurance fitness on a treadmill or behavior in an open field [2,3,4,5,6,7]. As such, both FLs represent a model that encompasses the diversity of high-fertility phenotypes. Together with the breeding for increased fertility, some unintended effects have also been observed. As an example, both genders of the fertility lines showed increased body weight at the time of mating. This effect is more pronounced in FL2 females, while FL1s reach the same body masses as the male animals [2]. As the selection is aimed toward large litters, this can suggest that an overall growth in body weight of the dams might be cooperative for proper gestation/maintaining pregnancy, but there is no evidence suggesting that the increase in body size also means higher body weight or higher fat content, which is usually associated with fertility issues. The literature usually links altered reproductive cycles and hormonal dysfunctions with altered ovulation processes in phenotypes of decreased fertility [8,9,10,11,12,13]. However, our recent study showed how FLs manage to reach the goal of increased offspring and ovulation rates even with hormonal misbalances, disruption of the reproductive cycle, and, sometimes, lower pregnancy rates. Indeed, our experiments demonstrated a correlation between the elongation of some stages of the estrous cycle with higher levels of insulin, leptin, or glucagon (depending on the line) or with an increased time to delivery or lower pregnancy rates [14]. Furthermore, we demonstrated that the ovaries of these superfertile mouse lines contain an elevated number of high-quality oocytes rather than an increased pool of oocytes. Hence, the number of high-quality oocytes determines the increased ovulation rates and consequently the number of offspring, rather than the number of follicles per ovary [15].

Here, we aimed to evaluate whether a connection between hormonal imbalances and an increased number of ovulated oocytes exists, and whether it is correlated with the body size or weight of FL mice compared to control mice. To do so, we mainly focused on the analysis of the body and liver dimension, quantity of adipose tissue in the abdomen and around the ovary and fat content of different organs, from plasma and liver, and adipose tissue. Moreover, we focused on the lipid microenvironment of the ovary determining the fatty acid composition of follicles and oocytes of different qualities.

### 1.2. Lipids and Oocytes

Lipids not only represent an important source of nutrients for oocytes and embryos but also have crucial roles in transport, cell proliferation, functions of biological membranes, etc. Approximately 12.5% of the mass of the mouse embryo cytoplasm is composed of this type of organic molecule [16,17]. To the best of our knowledge, only a few studies have been reported concerning lipid analysis and fatty acid composition in different species of oocytes [18,19,20,21,22]. Fatty acid composition varies from one tissue to another, but in oocytes as in adipose cells, their main function is as an energy source. It has been recently described that the increased concentration of free fatty acids in blood, and simultaneously in follicular fluid, may affect cumulus–oocyte complex (COC) morphology and embryo quality [18]. Interestingly, high levels of saturated fatty acids, such as palmitic acid or stearic acid, have lipotoxic effects, but an increase in unsaturated fatty acids, such as oleic acid, has the opposite effect, preventing lipotoxicity [23]. The result of the exposure to saturated fatty acids (SFAs) during IVM of oocytes is a decrease in developmental competence of the future embryo, which could be reversed if oocytes are exposed to unsaturated fatty acids. In that way, a vital function is carried out by cumulus cells directly connected to the oocyte, together with nearby granulosa cells, which protect the oocyte against lipotoxicity. In fact, when oocytes were exposed to SFAs during IVM, all cumulus and granulosa cells protected the oocyte by incorporating fatty acids from the medium [24,25,26,27]. Unsaturated fatty acids are divided into monounsaturated and polyunsaturated fatty acids (MUFAs and PUFAs). Of primary importance are long-chain n-3 and n-6 PUFAs because they are part of the phospholipidic bilayer of cellular membranes and represent a scaffold for integral proteins, such as junctions between the oocyte and granulosa cells, among others, which makes them an important link for fertility [28,29]. Jungheim et al. (2013) studied the correlation between serum PUFAs and pregnancy in women undergoing IVF. They found that an increased n-6/n-3 PUFA ratio, and in particular LA/ALA (linoleic acid/α-linoleic acid) and AA/EPA (arachidonic acid/eicosapentaenoic acid) ratio, enhances inflammation, which in turn encourages prostaglandin (PG) synthesis and endometrial reception, increasing implantation and pregnancy rates [30]. One of the most important steps of oocyte maturation is meiotic resumption, and both granulosa/cumulus cells together with fatty acid oxidation play a primary role in this mechanism [31]. Investigating the composition of human fertilization-failed oocytes, Matorras et al. reported higher levels of SFAs, such as stearic and palmitic acid, and low levels of MUFAs and PUFAs [19]. To our knowledge, there are no studies concerning the fatty acid composition of mouse oocytes. However, our study can be helpful for a better understanding of the characteristics of the gonads of high-fertility lines, where a particular lipid composition can help the growth of a higher number of oocytes with good quality.

The aim of the present study was to analyze the fatty acid composition of several parts of the female mouse body, from abdominal and ovarian fat to plasma, liver and gonads, using capillary gas chromatography and investigating the n-6/n-3, LA/ALA and AA/EPA ratios. Within the ovaries, we compared fatty acid composition in small follicles, high-quality COCs, oocytes without cumulus, degraded oocytes, and granulosa cells coming from antral follicles to focus on the differences between the microenvironment of good- versus bad-quality oocytes. In addition, we examined the differences in FAs composition connecting those with the phenotypical characteristics of the mice. The “unique mouse lines” object of the study represents an important point for the entire work: we compared two high-fertility lines (Dummerstorf high-fertility lines, FL1 and FL2) and one high-protein content line (Dummerstorf DU6P line) with our unselected control mice.

We report that in our lines, the phenotype of higher fertility can coexist with increased fat weight and higher abdominal fat content. Moreover, an increased ovulation rate and litter size have a connection with higher ovarian fat weight but lower lipid content, together with a different fatty acid panel in plasma, granulosa cells and COCs.

We aim to continue the investigation to analyze the effect of some particular fatty acids during in vitro maturation/fertilization (IVM/IVF) processes in the control line to determine whether the addition of specific fatty acids to the medium can be confined to granulosa cells or oocytes to improve their maturation.

## 2. Results

### 2.1. The FL2 Line Is Larger Than FL1, but FL1 Line Grows Faster

Figure 1A summarizes the body weight of the mice at 42–60 and 75 days old. Comparing selection lines with the unselected control mice at the same age (Figure 1A), it has been found that FL1 females have a similar weight as control mice from 42 to 60 and 75 days. FL2 mice, instead, were heavier than control and FL1 females at every time point. Interestingly, comparing the mice in terms of weight gain (Figure 1B, percentage of weight gain at 60 and 75 days old, in relation to their weight at 42 days old), we obtained the opposite result, as control and FL2 lines seem to have a similar weight gain, but FL1 mice show a higher percentage of growth compared to the control. Concerning the DU6P line, even if those mice are the largest in dimension and the ones with the higher weight at every age, their weight gain is actually not different from that of FL1s, which means that even if from a different starting point, both FL1 and DU6P grow more than control mice.

### 2.2. Both Fertility Lines Show a Higher Quantity of Body Fat Than Control and DU6P Mice, and Their Abdominal Fat Has a Higher Content of Lipids, but Their Ovarian Fat Has Less

Figure 2A,D describes the significant increase in the quantity of abdominal and ovarian fat in both fertility lines, not only compared to control but also to DU6P mice, the ones with the largest dimensions. Calculating the amount of fat in relation to the body weight of the mice (Figure 2B,E), the differences remained significant, with both FLs showing more than double the amount of fat in comparison with the control. Indeed, in the abdomen, 1% to 1.5% of the total body weight is made up of fat in FLs vs. 0.5% in the control, and in the ovary, 0.4% to 0.5% in FLs vs. 0.2% in the control. Even if the amount of fat in absolute value is higher in DU6P compared to the control, the quantity of fat in relation to the body weight reaches the value found in control mice. In addition, we analyzed the lipid portion of abdominal and ovarian fat (Figure 2C,F), and we found significant differences between the lines and the type of fat. In the abdomen (Figure 2C), the increased amount of fat corresponded to an increased fat content in all the selected lines, consisting of a higher lipid proportion (approximately 30 to 40% on average in FLs and DU6P, respectively) compared to controls (approximately 20%). In the ovary (Figure 2F), the increase in fat amount corresponds instead to a lower lipid content, at least in FL1 (~33%) and DU6P (~31%) compared to the control (~38%), as well as in DU6P compared and FL2 (~36%). Hence, it seems that the abdominal fat contains more lipids than the ovarian fat in the selected lines, and the opposite occurs in the control.

### 2.3. FL1 Females Have the Largest Liver Related to Their Dimension

In our selected lines, the weight of the liver seemed to be significantly higher compared to in the control (Figure 3A), but the analysis of the liver weight in relation to the body weight (Figure 3B) clarified that the higher values in FL2 and DU6P primarily depended on their higher body weight or larger dimensions. Interestingly, in FL1 mice, the increase was not due to the higher body weight; indeed, the ratio value remained significantly higher than that in all the other lines, reaching 6% of the entire body weight on average. In addition, analyzing the fat content of liver samples (Figure 3C), we found no significant differences between the lines, which means that the lipid portion remains similar (approximately 4% to 5% of the liver is made up of fat). However, as in FL1 the weight of the liver is higher, we can speculate that the quantity of fat in their liver is increased, compared to the other lines, in absolute value.

We also analyzed the fat content in plasma samples and found that only in the FL1 line the lipid portion was lower than that in the control (Figure 4).

### 2.4. Levels of Total SFAs, MUFAs and PUFAs in All Tissues and the n-6/n-3 PUFA Ratio

The portion of SFAs in the samples coming from the ovary (small follicles, COCs, oocytes and granulosa cells) was significantly higher than that in plasma, liver and fat (Figure 5B: 80–95% on average vs. Figure 5A: 25–40%). The proportions of MUFAs and PUFAs also showed significant differences (Figure 5C–F). PUFAs represent the most abundant portion of FAs in plasma (45% to 50% depending on the lines, Figure 5E—Plasma). Simultaneously, we measured significantly lower plasma levels of total SFAs (29–31%) in all the lines compared to the control (34%) and higher levels of some corresponding MUFAs (mostly in DU6P, ~25% vs. 19% in control) and PUFAs (Figure 5A,C,E—Plasma).

The liver is the primary organ of de novo synthesis of fatty acids in mice. Here, and in fat samples, the lines followed the same tendency, with higher levels of MUFAs and lower levels of PUFAs in FL1 and DU6P compared to the control, even if fat samples showed increased levels of MUFAs and lower SFAs compared to the liver in all the lines (Figure 5A,C,E).

Figure 5B,D,F shows the fatty acid composition of the ovarian microenvironment. As expected, we did not detect significant differences between the lines in low-quality oocytes without cumulus (Figure 5B,D,E—Oocytes Woc). Only in control mice did we find the same FA composition in high-quality cumulus–oocyte complexes (Figure 5B,D,E—COCs), as well as in granulosa cells (Figure 5B,D,E—Granulosa cells) and degraded oocytes (Figure 5B,D,E—Degraded oocytes), comprising ~90% SFAs, 5% MUFAs and 5% PUFAs. In granulosa cell samples, we detected the largest difference within the ovarian microenvironment between the lines, with lower SFAs and higher PUFAs and/or MUFAs in FL1 and DU6P. The FL2 line followed a similar tendency, but there was no significant difference compared to the control.

In addition, we calculated the ratio between n-6/n-3 PUFAs (Figure 6A). The lines seem to show the same tendency, not only in fat samples but also in COCs, with a significant increase in FL1 COCs and oocytes without cumulus (WoC) compared to the control. Moreover, the ratio of LA/ALA (Figure 6B) was significantly higher in FL2 plasma and liver, but the AA/EPA ratio (Figure 6C) was significantly higher in DU6P plasma, lower in FL1 liver and higher in ovarian fat (but not abdominal fat) in both lines.

### 2.5. Fatty Acids in Plasma

Shifting from the general composition to the individual FAs per sample type, in plasma (Figure 7), the most abundant fatty acids in the control line were linoleic acid (LA, Figure 7C) and palmitic acid (PA, Figure 7A), but in selection lines, we observed a slight decrease in LA and PA levels, which was significant in DU6P. Those decreases translate into higher levels of oleic acid (OA, Figure 7B) in DU6P; stearic acid (SA) and OA in FL1 (not significant); arachidonic acid (AA, Figure 7C) and DHA (Figure 7D) in FL2; plus a slight increase in EPA (C20:5 n-3, not shown).

### 2.6. Fatty Acids in Liver

The most noticeable difference between the lines in liver is represented by higher levels of the MUFA OA in FL1 and DU6P compared to control (Figure 8B), which translates into lower levels of PUFAs DHA in FL1, AA in FL1 and DU6P, and LA in DU6P (Figure 8C), as well as decreasing SFA SA in FL1 and PA in DU6P (Figure 8A). In addition, we found a decrease in LA in FL2 cells (Figure 8C).

### 2.7. Fatty Acids in Abdominal and Ovarian Fat

The adipose tissue taken from the abdomen showed the same increase in OA in FL1 and DU6P (Figure 9B), which in both lines translates into lower levels of the PUFA LA (Figure 9D). We also detected differences in other MUFAs (Figure 9C), with higher palmitoleic acid (POA) in FL1 and lower POA in DU6P compared to the control, and an increase in cis-vaccenic acid in FL2 and DU6P. In general, the amount of total SFAs was similar in all the lines (Figure 5A and Figure 9A), with a difference in total MUFAs (from 35–36% in control and FL2 to 38–41% in FL1 and DU6P depending on the organ, Figure 5C and Figure 9B,C) and PUFAs (37–39% in control and FL2 vs. 34–35% in FL1 and DU6P, Figure 5E and Figure 9D). In addition, we tried to highlight some particularities in FLs that could explain the increased fat content and the connection with higher-fertility traits. We found only one FA with the same trend in FL ovarian fat compared to control and DU6P, which is the SA (significantly lower in FLs (~2.5%) compared to control (~3.9%), data not shown). The same pattern was visible in the abdominal fat (Figure 9A, not significant).

### 2.8. Fatty Acids in Follicles, COCs, Oocytes without Cumulus Cells and Granulosa Cells

The fatty acid composition in the samples from the ovaries (Figure 5B) was very different from that in the plasma, liver and fat (Figure 5A). Here, levels of SFAs were significantly higher (from 80 to 95% depending on the sample type and line) compared to the other tissues and, in general, there were higher PUFA levels in oocytes without cumulus compared to small follicles, COCs and degraded oocytes (Figure 5F). In follicles, the most conspicuous change between the lines was represented by higher levels of PUFAs (Figure 5F—follicles, not significant) in FL2 (4.4%) and DU6P (3.8%) compared to control and FL1 (2 and 2.5%). An approximately 1% increase was also shown in the levels of MUFAs in the same lines (Figure 5D, not significant). On the other hand, even if control and FL1 show higher levels of total SFAs (an approximately 3% increase compared to the other two lines, Figure 5B), we noticed some slight increases in all the lines depending on the acid: C12:0 in FL2, C14:0 and C16:0 in FL1, C24:0 and C26:0 in DU6P. Almost all the others are higher in the control (not shown).

The total amount of SFAs in cumulus–oocyte complexes (COCs, Figure 5B and Figure 10E,F,G) decreased in favor of MUFA and PUFA portions in all the selected lines (Figure 5F). Again, we found the same 3–4% difference in SFAs (higher in control and FL1) and MUFAs (higher in FL2 and DU6P), with a perceptible difference in PUFAs (higher in FL2 and DU6P).

We did not find significant differences in oocytes without cumulus (WoC, Figure 5B,D,F) between the lines, but we noticed a significant increase in PUFAs (Figure 5F ~7–8%) compared to COCs (4–5%) and small follicles (2–4%), together with a similar decrease in SFAs (Figure 5B).

Degraded oocytes are those with the lowest quality. In control and FL2 lines, they contain more PA and SA; higher C22:0 and C20:0 are shown in FL1 and DU6P, and C14:0 increased in all the selected lines (data not shown). Levels of PUFAs are increased only in DU6P (6.4%, Figure 5F) compared to control (3.6%) and FL1 (3.3%). A perceptible increase was also observed in FL2 (4.6%).

Granulosa cells usually protect the oocyte by incorporating “bad” fatty acids from the medium during IVM-IVF processes. Here, we found the largest differences between the lines within all the “small tissues/samples” (Figure 5B,D,F). Higher levels of total SFAs were observed only in control and FL2 (mostly C22:0, C24:0, C26:0, Figure 10C,D), and higher levels of MUFAs (mostly OA) and PUFAs (mostly n-6) were observed in FL1 and DU6P (not shown).

A relevant part of this comparison in small tissue concerns SFAs from C20 to C26 (Figure 10) in FL1. Some of them seemed to be lower in granulosa cells (Figure 10C,D) and oocytes without cumulus (Figure 10A,B) but higher in COCs (Figure 10D,E), which suggests that in COCs, they are concentrated inside the oocyte and not in granulosa cells around it (Figure 10). Indeed, granulosa cells in the FL1 line showed significant increases in the corresponding PUFAs (Figure 5B). Slight decreases in the same SFAs were found in follicles (not shown).

For an overview of the fatty acid composition, all together in all the types of samples in the four lines, we created heatmaps (Figure 11A) and principal-component analysis graphs (PCAs, Figure 11B), where mice mostly segregate per line, and the FL1 line seems to be the most different line. Five FL2 mice seem to be similar in FA composition to the control, as they segregate together; all the others segregate with DU6P’s, but they are always different from FL1.

## 3. Discussion

Worldwide human fertility rates have decreased in recent decades due to several socioeconomic reasons [32]. This leads to increasing IVF cycles, which are prone to select for “low-quality genes”. Moreover, the main knowledge on reproduction is based on adverse fertility effects, and “high-fertility genes” are rarely sought, as 99% of mouse models showing a reproductive phenotype are knockout and describe subfertility or infertility [33,34]. Dummerstorf high-fertility mouse lines were selected for increased litter size for more than 190 generations; hence, these worldwide unique mouse lines can provide some insight into the desired phenotype of high fertility. After more than 50 years of selection, both fertility lines doubled the number of pups per litter from approximately 11.5/litter in the control to approximately 20.6 and 21.4 for FL1 and FL2, respectively, due to an increase in their ovulation rates [2,3,5,6]. Females of Dummerstorf lines managed to reach this goal of higher ovulation rates, increasing the number of high-quality oocytes per ovary compared to the control [15], even showing alterations in some metabolic hormones, such as leptin, insulin, and glucagon, connected with their nonconventional reproductive cycle [14].

The literature describes disaccording opinions about fat content or overweight and lower/higher fertility [12,35,36,37], which we analyzed in our superior fertile mouse lines. The focus of the present study was to investigate whether the hormonal alterations found in fertility lines can translate into a higher body fat content compared to the control and how it can be connected with their higher ovulation rate. We found that both FLs showed increased abdominal and ovarian fat content, not only in comparison with the control but also to the DU6P line (Figure 2), which is characterized by the largest dimension and protein content and a significantly higher number of follicles per ovary compared to controls and FLs, but also a lower quality of their oocytes [15]. In addition, FL1 revealed an unexpected increase in the weight of the liver relative to body weight (Figure 3). To gain insight into lipid metabolism, we studied the fatty acid composition in plasma, liver, fat, follicles, COCs, lower-quality oocytes and granulosa cells of our mice and correlated the differences between the lines with their phenotypical characteristics. In all the selected lines examined in our study, plasma levels of saturated FAs (SFAs) were significantly lower than in the control, with a corresponding increase in MUFAs in DU6P and unsaturated fatty acids (MUFAs and PUFAs together) in FLs (Figure 5). We speculate that they manage to reach large litter sizes even presenting higher fat weight and/or content controlling/balancing the levels of unsaturated fatty acids at the expense of SFAs, as increased levels of unsaturated FAs likely prevent lipotoxicity caused by SFAs during oocyte/embryo development [23]. High-fertility lines might reach this goal thanks to a different quantity of the conversion enzymes or their expression level or post-transcriptional modifications, which we aim to further analyze. Heatmap analysis revealed that mice within the same line mostly clustered together, and FL1 was the most different, as it segregated by itself on almost all levels (more heatmaps of FA analysis in single tissues can be provided on request). Additionally, within the lines, we found that several mice with similar phenotypical traits, different from other specimens of the same line, segregate together as they have similar FA compositions. Thus, the lines can be clustered solely by their lipid pattern, which becomes informative for the next levels of analysis. The fatty acid panel in the liver analyzed by heatmap and PCA (provided on request) again shows clusters by line, with FL1 revealing the largest differences, including higher OA and a lower AA, DHA, and AA/EPA ratio. As higher levels of SFAs can cause more severe oxidative stress but higher levels of unsaturated FAs cause serious lipid accumulation [38], this could play a role in their hepatomegaly together with the higher plasma levels of insulin found earlier [14,35,39]. Moreover, the FL1 line is the only one with lower fat content in plasma than the control, which could be one of the reasons why it is a healthy mouse line, even showing an enlarged liver. With this purpose, we would like to further analyze the expression levels of some conversion enzymes, such as elongases and/or desaturases in FLs, which convert saturated to unsaturated FAs and vice versa, to achieve some more deep evaluations of their way of managing the healthy fertility traits. Indeed, the linear proportional trend of increase/decrease between saturated/unsaturated FAs (e.g., SA/OA/LA, etc.) in different organs or within the same sample could probably be explained enzymatically [40,41]. In oocytes either without cumulus or degraded (lower quality), the differences between the four lines were very low, as expected (Figure 5). In the liver, abdominal fat, ovarian fat and granulosa cells, we found more similarities between control/FL2 and FL1/DU6P, which is consistent with the higher insulin levels found in DU6P and FL1. We should take into account that DU6P is the line with the highest quantity of oocytes that lose the cumulus of granulosa cells during follicular growth [15], which translates into the fact that those cells could lose the characteristic of “helper cells” [25,26,27]. Moreover, FL1 mice have the largest number of poly-ovulatory follicles (POFs), which means that two or more oocytes share the same follicle and are connected with the same granulosa cells. In addition, those oocytes can reveal similar or different quality [15]. Hence, these reflections might explain the insistent presence of PUFAs in granulosa cells of DU6P and FL1. In Cumulus–oocyte complexes, the similarities are switched to control/FL1 and DU6P/FL2 (Figure 5), which could fit with their similarities in estrous cycle length, even if we did not find specific FAs/line connections. Higher plasma levels of leptin have been described in FL2 mice in comparison to the other lines [14]. Leptin is linked with increased fat intake, but it is also a modulator of oocyte maturation, as higher leptin levels lead to increased rates of GVBD (germinal vesicle breakdown) and decreased rates of atresia [42]. In our opinion, this could help to explain the differences between FL1 and FL2 in terms of fat content (Figure 2), but also the presence of POFs [43] predominant in FL1 where the GVBD rate could be lower. In addition, we speculate that the presence of more than one oocyte per follicle in FL1 ovaries might explain the different panel of FAs in granulosa cells (Figure 6)., which have in those lines different roles, as more oocytes share the same family of granulosa cells in FL1, and in some cases those oocytes have different quality.

It was recently found that long-chain n-3 PUFA (ALA, EPA, DHA) but not n-6 PUFA (LA) can modulate or restore fertility indices in high-fat-diet mice fed with SFAs in terms of litter size, litter weight and sex ratio [44]. In addition, DHA can be reduced by body metabolism to generate vitamin C, which has been found to be able to revert the oxidative stress produced by free fatty acids [45] and to restore ovarian follicular reservation in old mice, improving the number of primordial, primary, secondary and antral follicles [46]. In our study, we found a significant decrease in SFAs in plasma (Figure 5), particularly PA, in both FLs (Figure 7A, significant only in DU6P). We also detected lower levels of n-6 LA, increased levels of AA (n-6) and EPA (n-3) in FL1, and AA and DHA (n-3) in FL2 (Figure 7), as well as higher LA/ALA in FL2, which can fit with their high litter size and reproductive performance. However, inside the ovary, the lipid fractions of FAs are different (Figure 10). This leads us to speculate that FAs could play an important role in creating a healthy microenvironment for the oocytes, in addition to the difference between the abdominal fat (rich of lipids, Figure 2C) and the adipose tissue surrounding the ovary (with less lipids and higher AA/EPA at least in FL1 line, Figure 2F and 6C). We would like to test some fatty acids, in particular SFAs from C20:0 to C26:0, in the control line, detecting the place in which they are stored within the ovary, to analyze the effects on their reproductive phenotype, e.g., maturation rates or high-quality oocyte rates.

## 4. Material and Methods

### 4.1. Mouse Lines, Diet and Housing

For our study, we used females of three Dummerstorf outbred selected mouse lines in comparison with the unselected control line.

High-fertility lines FL1 and FL2 were bred for the first 162 generations following the Dummerstorf selection index (1.6 × number of offspring + birth litter weight) followed by nine generations of best linear unbiased prediction (BLUP) breeding value estimation, focusing only on a higher number of pups per litter. After generation number 171, animals were bred according to the higher offspring per female [3]. Whereas FL1 females have not been treated, the estrus of the females in FL2 has been synchronized by application of the gestagen chlormadinone acetate up to the 23rd generation. For general breeding, one male and one female mouse (nine weeks old) were mated for 15 consecutive days. The generation sizes were kept between 60−100 breeding pairs per generation [1] to maintain the outbreed character; consequently, these outbred mouse lines are more heterogeneous and biodiverse in nature than classical inbred mouse lines.

DU6P, selected for 154 generations from a noninbred mouse model, according to the highest protein amount on the male side [47,48]. Previous studies have shown a coevolution of high protein and glycogen contents during selection experiments and have identified PTEN as a gatekeeper for muscle mass in middle-aged female DU6P mice [47]. We included the DU6P line in our study, as those females show a similar and irregular reproductive cycle, together with altered plasma levels of some metabolic hormones, such as insulin and ghrelin, connected with GnRH (gonadotropin-releasing hormone) secretion, as FL mice do [14]. Those mice interestingly show a higher quantity of follicles per ovary compared with the control and both fertility lines [15]. However, the quantity of high-quality oocytes is lower than in FLs, but a bit higher than in the control. Indeed, it follows the same slight increase found for the offspring rate per litter (13.7 in DU6P versus 11.5 in the control).

We used the unselected mice as a control line (control). All the animals were housed in specific pathogen-free areas in polysulfone cages that were 26.7 × 20.7 × 14 cm (Tecniplast, Germany—H-Temp PSU) and maintained under a normal 22.5 °C temperature and 50% humidity conditions with free access to water and food. Pellet sniff M-Z autoclavable concentrate (catalogue number V1124-300, Soest, Germany) is composed of 12% fat, 27% protein and 61% carbohydrates (nitrogen-free-extracts 51.2%). The mice were maintained/housed with a 12:12 h light–darkness cycle. The experiments were performed in accordance with both national and international guidelines and approved by our Animal Protection Board (APB) from the Research Institute for Farm Animal Biology in Dummerstorf, Germany.

### 4.2. Experimental Design and Sample Preparation

Forty-four female animals, eleven per line (control, FL1, FL2 and DU6P), were housed as explained above from day 42 to 75 after birth and weighed during the entire period at three time points: at 42, 60 and 75 days old. The weights of the three lines in the present study were compared with those of control females to calculate the differences in body weight at each time point, together with the body weight growing from the 42nd day to the 60th and 75th days after birth. All the mice (aged between 75 and 89 days) were euthanized by CO_2_ inhalation and immediately dissected at the dioestrus stage, as previously reported [14]. Briefly, vaginal secretions were collected, pipetting directly to the vaginal opening, and the estrous stage was determined by vaginal smear examination directly under the microscope without previous staining. The vena cava was cut, and the blood was extracted from the thoracic cavity, mixed with 0.5 M EDTA (pH 8.5) in an Eppendorf tube, and centrifuged to obtain plasma.

Adipose tissue from the abdomen and around the ovary, together with the entire liver, were extracted, weighed and stored at −80 °C for further fatty acid analysis. In addition to the real weight of those tissues, their weight in relation to the entire body weight was calculated in percentage (e.g., if the body weight is 40 g and the liver weights 2 g, then the relation corresponds to the 5% (2/40) of the body weight). The ovaries were cleaned from the surrounding fat and incubated in warm M2 medium, and their oocytes were immediately analyzed as described previously [15]. Briefly, we used two different protocols to analyze the entire pool of follicles per ovary (protocol 1), and only oocytes coming from antral follicles (protocol 2) and selected the oocytes depending on their quality.

After oocyte selection, the following samples were used for fatty acid analysis: per animal, approximately 30 small follicles, ~20 degraded oocytes, ~20 oocytes without cumulus, ~15 COCs and granulosa cells from the antral follicles (>300 µm).

### 4.3. Lipid Extraction and Transesterification

Nine types of samples per animal were used for lipid extraction and subsequent fatty acid analysis for a total of 396 samples (9 samples × 4 lines × 11 mice per line): plasma, liver, abdominal fat, ovarian fat, small follicles, COCs, oocytes without cumulus, granulosa cells and degraded oocytes with a granular cytoplasm. We used two different methods depending on the sample types.

#### 4.3.1. Liver, Abdominal and Ovarian Fat

The frozen liver and adipose tissue samples were cut into small pieces and homogenized. For lipid extraction, approx. one hundred milligrams of tissue was weighed in a tube. Each Precellys tube contained 20 pieces of 2.8 mm bulk beads and 2 pieces of 5 mm bulk beads (Zirconium oxide Precellys beads, Bertin Instruments Technologies, Montigny-le-Bretonneux, France). After the addition of 3 mL of methanol and nonadeconoic acid (19:0) as an internal standard, the extracts were homogenized three times at 25 sec intervals at 4 °C and 6500 rpm using a homogenizer (Precellys Evolution, Bertin Instruments Technologies, Montigny-le-Bretonneux, France). The homogenates were vortexed and transferred to Pyrex tubes (Pyrex, Hayes, UK) containing 8 mL of chloroform. After that, the Precellys tubes were washed two times with 1 mL of methanol and added to the Pyrex tubes. The details of lipid extraction were recently described [49].

#### 4.3.2. Plasma, Follicles, COCs, Oocytes without Cumulus Cells, Granulosa Cells and Degraded Cells

Plasma (approx. 0.5 mL) or follicles and different cells (approx. 0.1 mL) were defrosted and added dropwise to a tube containing 8 mL of chloroform/methanol (2:1, *v*/*v*) with 60 µL C19:0 as an internal standard (60 mg/mL) at room temperature. The plasma sample preparation has been described in detail elsewhere [50]. All solvents used for tissue, plasma, and cell lipid extraction contained 0.005% (*w*/*v*) t-butylhydroxytoluene (BHT) to prevent oxidation of PUFAs. After filtration, the lipid extracts of tissues, plasma and cell samples were stored at 5 °C for 18 h in the dark and subsequently washed with 0.02% CaCl_2_ solution. The organic phase was separated and dried with a mixture of Na_2_SO_4_ and K_2_CO_3_ (10:1, *w*/*w*), and the solvent was subsequently removed using a vacuum centrifuge (ScanSpeed 40; LaboGene, Allerød, Denmark) at 2000 rpm/min and 30 °C for 30 min. The lipid extracts were redissolved in 300 μL of toluene, and a 25 mg aliquot was used for methyl ester preparation [51]. Briefly, for transmethylation, 2 mL of 0.5 M sodium methoxide in methanol were added to the lipid extracts, which were shaken in a 60 °C water bath for 10 min. Subsequently, 1 mL of 14% boron trifluoride in methanol was added to the mixture, which was then shaken for an additional 10 min at 60 °C. The fatty acid methyl esters (FAMEs) were extracted twice with 2 mL of n-hexane and stored at −18 °C until use for high-resolution gas chromatography (HR-GC) analysis.

### 4.4. Fatty Acid Analysis

The fatty acid analysis of all sample extracts was performed using capillary GC with a CP-Sil 88 CB column (100 m × 0.25 mm, Agilent, Santa Clara, CA, USA) installed in a PerkinElmer gas chromatograph CLARUS 680 with a flame ionization detector and split injection (PerkinElmer Instruments, Waltham, MA, USA) as described earlier [50]. Briefly, hydrogen was used as the carrier gas at a flow rate of 1 mL × min^−1^, while the split ratio was 1:20, with the injector and detector set at 260 and 280 °C, respectively. The GC oven temperature program was 150 °C for 5 min, followed by a heating rate of 2°/min until 200 °C and then being kept for 10 min, followed by a final heating rate of 1°/min until 225 °C and then being kept for 20 min. For the calibration, the reference standard mixture “Sigma FAME” (Sigma–Aldrich, Deisenhofen, Germany), the methyl ester of C18:1cis-11, C22:5n-3, and C18:2cis-9,trans-11 (Matreya, State College, PA, USA), C22:4n-6 (Sigma–Aldrich, Deisenhofen, Germany), and C18:4n-3 (Larodan, Limhamn, Sweden) were used. The five-point calibration of single fatty acids ranged between 16 and 415 µg/mL and was assessed after GC analysis of five samples. Fatty acid proportions were displayed as individual percent of total fatty acids.

### 4.5. Methods of Analysis of Fatty Acids

We analyzed the levels of total SFAs, total MUFAs and total PUFAs in all sample types (plasma, liver, abdominal fat, ovarian fat, small follicles, COCs, oocytes without cumulus, degraded oocytes and granulosa cells coming from antral follicles), including the ratios between total n-6/total n-3, LA:ALA and AA:EPA in some of them. However, for the sake of clarity, only the most abundant fatty acids are shown in the results section per sample type; the composition of all matrices in all the lines can be found in the Appendix A. For a complete overview, we created a heatmap and a principal component analysis (PCA) of the complete composition of FAs in all the tissues.

### 4.6. Statistical Analysis

Statistical significance was analyzed using GraphPad Prism 5 (GraphPad Software, San Diego, CA, USA) by one-way ANOVA followed by Bonferroni’s or Tukey’s posttest. Data are illustrated as the mean, ± SEM (standard error of the mean) and designated significant if *p* < 0.05 (Bonferroni: * *p* < 0.05; ** *p* < 0.01; *** *p* < 0.005; Tukey: different letters correspond to different levels, with *p* < 0.05). Heatmap and PCA were realized using the Clustvis web tool [52].

## 5. Conclusions

In conclusion, the deep analysis of our fertility lines can become a very informative way to obtain insights into the “physiological pathways” they have been using during the selection period to avoid their adverse characteristics having an impact on their higher-fertility traits. For example, higher insulin, leptin or glucagon levels in plasma explain the unusual estrous cycle length but might also help them to achieve an increased amount of fat, or a higher fat content, which in turn supports the follicular growth and better ovarian quality. In addition, we speculate that the difference between the adipose tissue in the abdomen and that around the ovary might explain their ability to achieve more pups per litter without any retardation in the offspring population compared to control. Moreover, the differences found in the fatty acid composition in different organs and different quality oocytes help us to form some ideas concerning the next steps of our research that comprehend the deep analysis of the effect of particular FAs in control mice, even during IVM of immature oocytes, or in addition to their diet, to increase their ovarian quality. These discoveries can be highly helpful not only for increasing the production of farm animals, but also for solving some fertility issues and increasing fertility rates in humans showing similar characteristics.

## Figures and Tables

**Figure 1 ijms-23-10245-f001:**
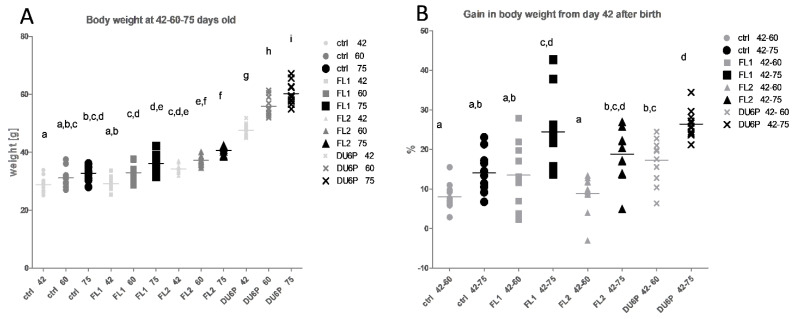
(**A**) Mouse weights in grams in control (circles), FL1 (squares), FL2 (triangles) and DU6P (crosses) at 42–60 and 75 days old. Each line is represented using the same symbol, and each age in different tonalities and dimensions, starting from the lightest/smallest (42 days old) and finishing with the darkest/largest (75 days old). (**B**) Gain in body weight in control (circles), FL1 (squares), FL2 (triangles) and DU6P (crosses) from 42 days old to 60 days old (clear color) and from 42 to 75 days old (dark color). Data are illustrated as the mean ± SEM (standard error of the mean) and designated significant if *p* < 0.5 (Tukey posttest: different letters correspond to different levels of significance).

**Figure 2 ijms-23-10245-f002:**
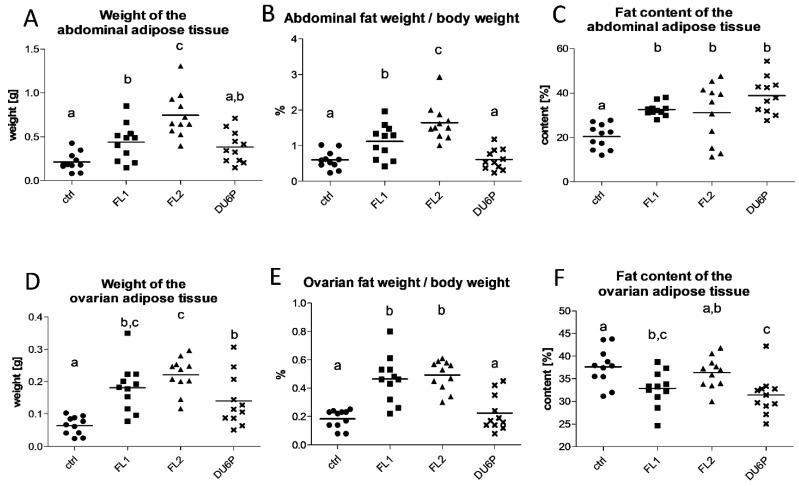
Weight of the adipose tissue coming from abdomen (**A**) and around the ovaries (**D**), weight of the adipose tissue, with respect to the body weight, in the abdomen (**B**) and ovaries (**E**) in control (circles), FL1 (squares), FL2 (triangles) and DU6P (crosses) lines. Fat content/lipid portion of the adipose tissue of the abdomen (**C**) and around the ovaries (**F**) in the same lines. Data are illustrated as the mean ± SEM (standard error of the mean) and designated significant if *p* < 0.05 (Tukey posttest: different letters correspond to different levels of significance).

**Figure 3 ijms-23-10245-f003:**
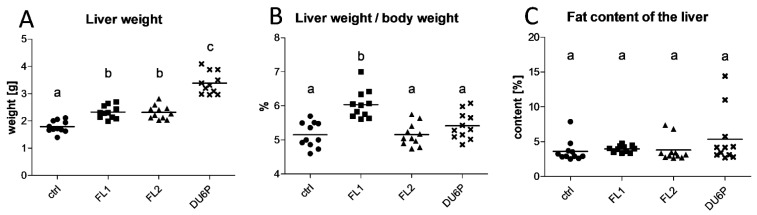
Liver weight (**A**) and liver weight ratio with respect to body weight (**B**) in control (circles), FL1 (squares), FL2 (triangles) and DU6P (crosses) lines. Content of fat in the liver in the same lines (**C**). Data are illustrated as the mean ± SEM (standard error of the mean) and designated significant if *p* < 0.05 (Tukey posttest: different letters correspond to different levels of significance).

**Figure 4 ijms-23-10245-f004:**
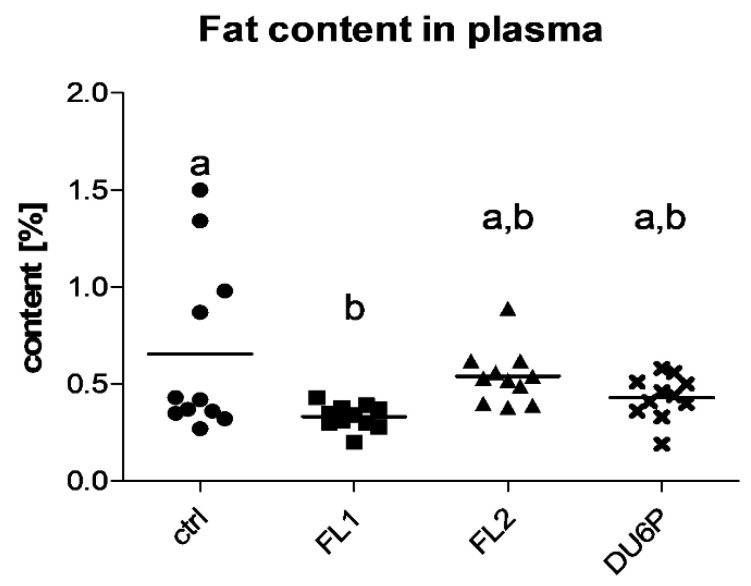
Fat content in plasma samples in control (circles), FL1 (squares), FL2 (triangles) and DU6P (crosses). Data are illustrated as the mean ± SEM (standard error of the mean) and designated significant if *p* < 0.05 (Tukey posttest: different letters correspond to different levels of significance).

**Figure 5 ijms-23-10245-f005:**
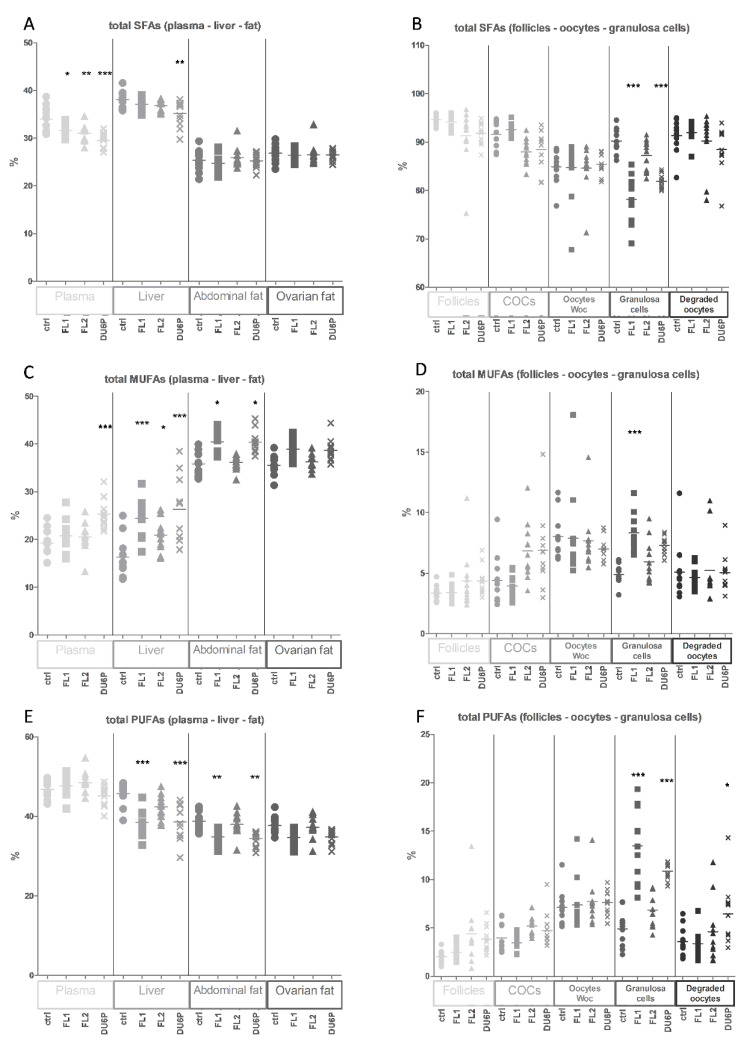
Fatty acid composition in control (circles), FL1 (squares), FL2 (triangles) and DU6P (crosses). (**A**,**C**,**F**) (large symbols) show the composition of SFAs (**A**), MUFAs (**C**) and PUFAs (**E**) in plasma, liver, abdominal and ovarian fat. (**B**–**D**) (small symbols) show the composition of SFAs (**B**), MUFAs (**D**) and PUFAs (**F**) in follicles, COCs (Cumulus–oocyte complex), oocytes WoC (without cumulus), granulosa cells and degraded oocytes. Different tissues are represented using different tonalities. Data are illustrated as the mean ± SEM (standard error of the mean) and designated significant if *p* < 0.05 (Bonferroni posttest: * *p* < 0.05; ** *p* < 0.01; *** *p* < 0.005, in comparison with control of the same group).

**Figure 6 ijms-23-10245-f006:**
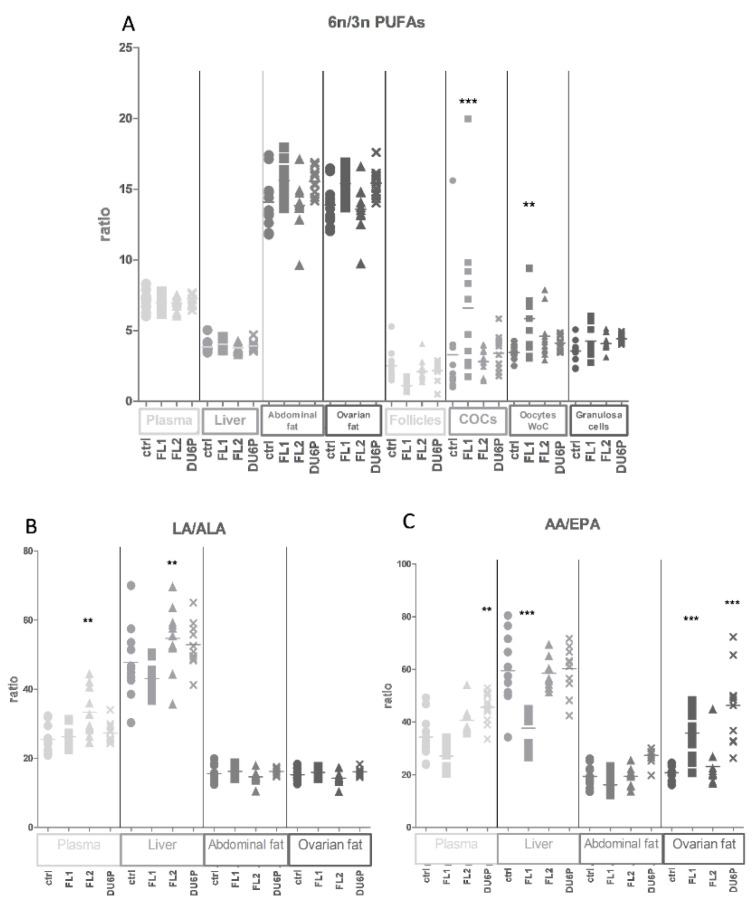
Fatty acid composition in control (circles), FL1 (squares), FL2 (triangles) and DU6P (crosses). (**A**) shows the PUFA n-6/n-3 ratio in plasma, liver, abdominal and ovarian fat, follicles, COCs, oocytes WoC (without cumulus) and granulosa cells. (**B**,**C**) show the ratio between the n-6 LA and the n-3 ALA (**B**) and the n-6 AA and the n-3 EPA (**C**) in plasma, liver and fat. Different tissues are represented using different tonalities and dimensions. Data are illustrated as the mean ± SEM (standard error of the mean) and designated significant if *p* < 0.05 (Bonferroni posttest: ** *p* < 0.01; *** *p* < 0.005, in comparison with control of the same group).

**Figure 7 ijms-23-10245-f007:**
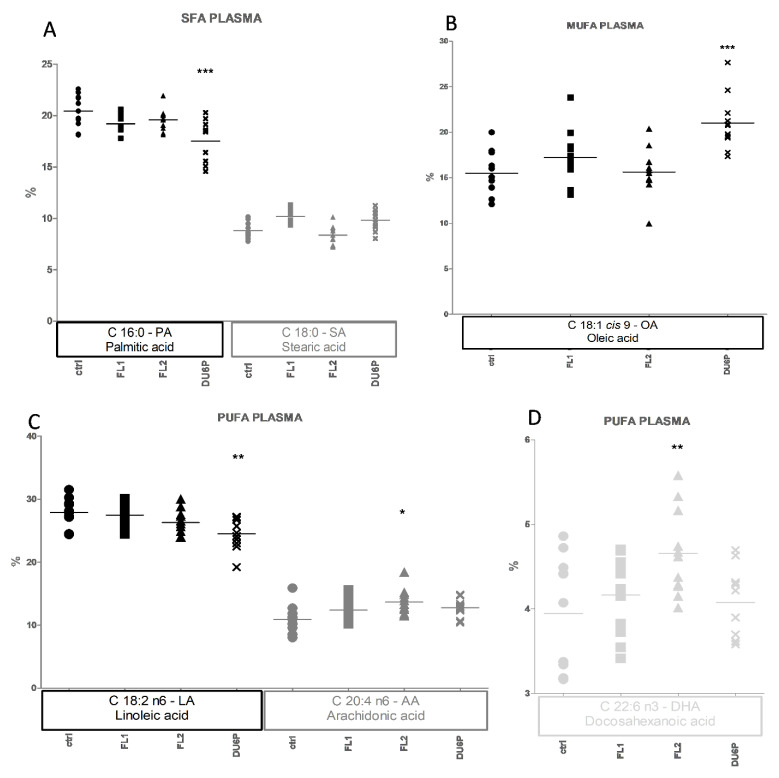
Plasma composition of the most abundant SFAs (small symbols, **A**), MUFAs (medium symbols, **B**), and PUFAs (large symbols, **C**,**D**) in control (circles), FL1 (squares), FL2 (triangles) and DU6P (crosses). Different FAs are represented using different tonalities. Data are illustrated as the mean ± SEM (standard error of the mean) and designated significant if *p* < 0.05 (Bonferroni posttest: * *p* < 0.05; ** *p* < 0.01; *** *p* < 0.005, in comparison with control of the same group).

**Figure 8 ijms-23-10245-f008:**
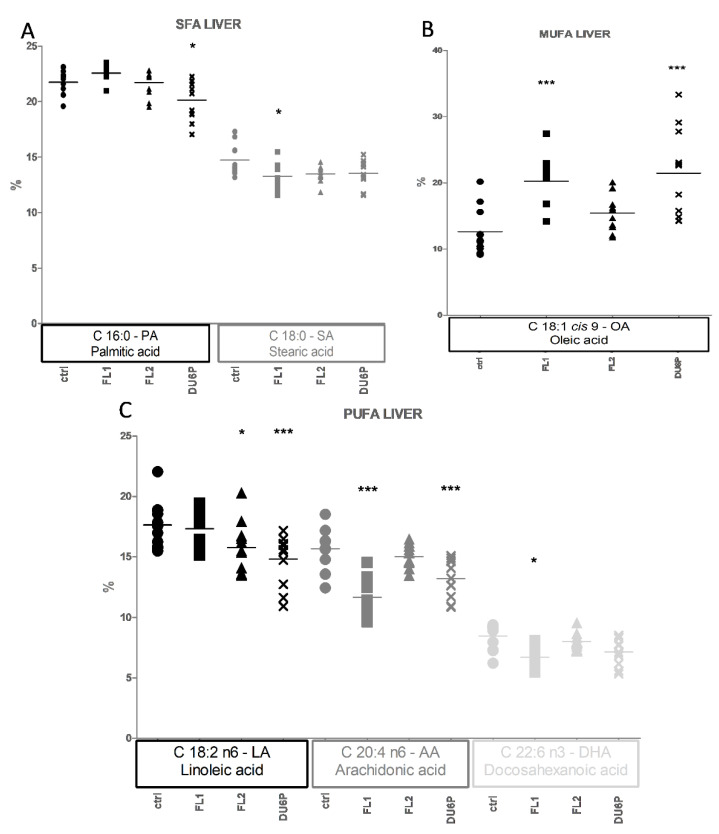
Liver composition of the most abundant SFAs (small symbols, **A**), MUFAs (medium symbols, **B**), and PUFAs (large symbols, **C**) in control (circles), FL1 (squares), FL2 (triangles) and DU6P (crosses). Different FAs are represented using different tonalities. Data are illustrated as the mean ± SEM (standard error of the mean) and designated significant if *p* < 0.05 (Bonferroni posttest: * *p* < 0.05; *** *p* < 0.005, in comparison with control of the same group).

**Figure 9 ijms-23-10245-f009:**
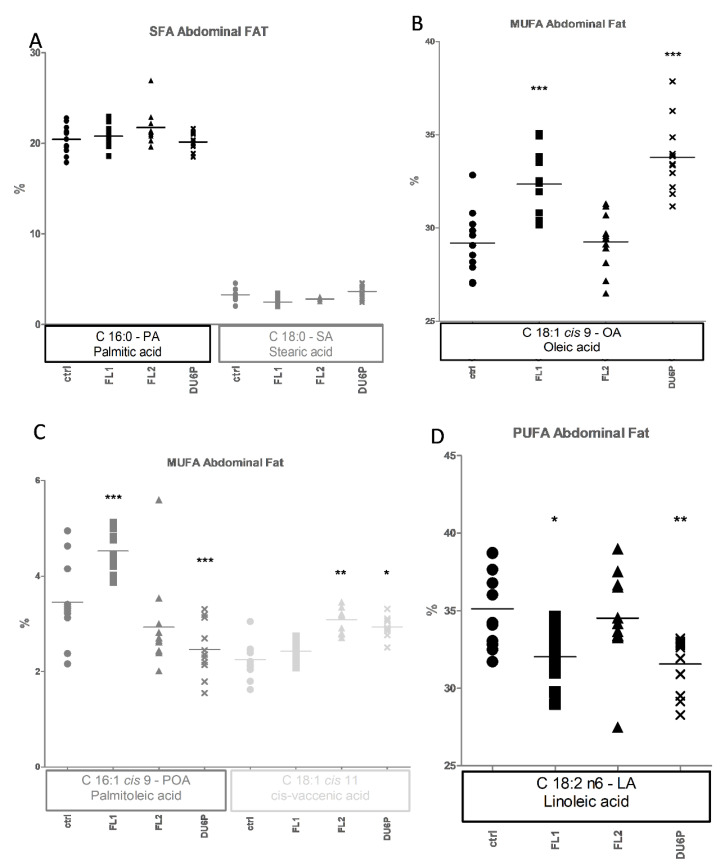
Abdominal fat composition of the most abundant SFAs (small symbols, **A**), MUFAs (medium symbols, **B**,**C**), and PUFAs (large symbols, **D**) in control (circles), FL1 (squares), FL2 (triangles) and DU6P (crosses). Different FAs are represented using different tonalities. Data are illustrated as the mean ± SEM (standard error of the mean) and designated significant if *p* < 0.05 (Bonferroni posttest: * *p* < 0.05; ** *p* < 0.01; *** *p* < 0.005, in comparison with control of the same group).

**Figure 10 ijms-23-10245-f010:**
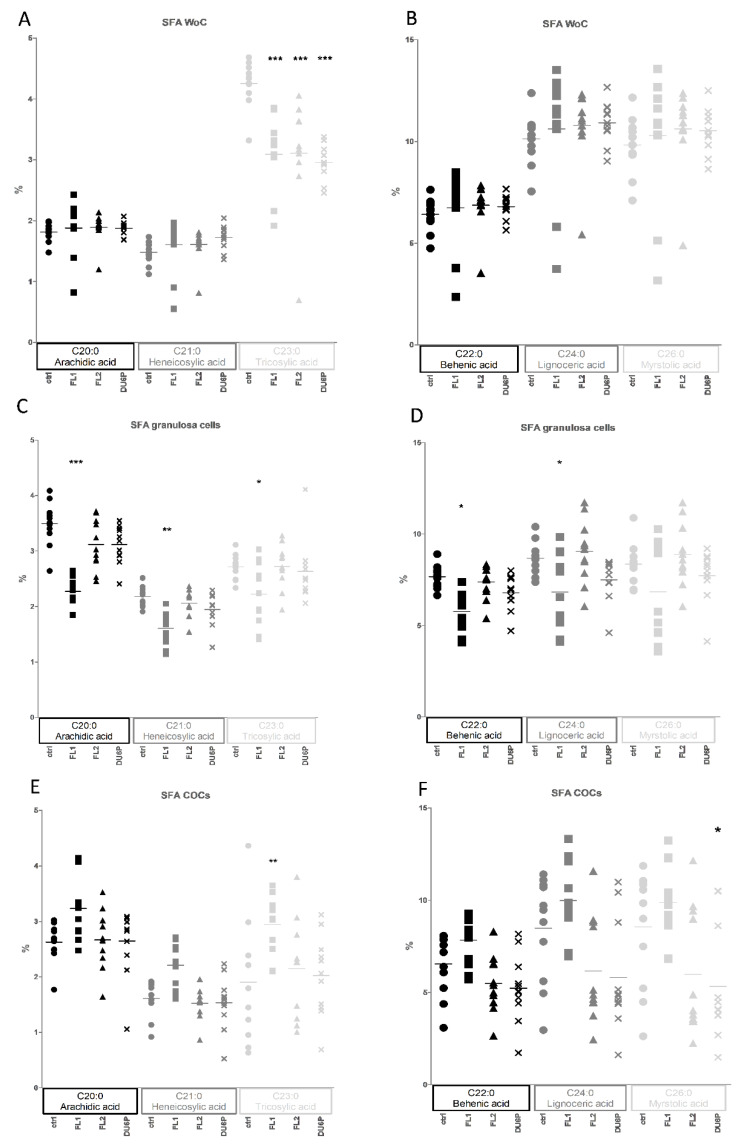
Percentages of the most abundant SFAs in oocytes without cumulus (**A**,**B**), granulosa cells (**C**,**D**) and COCs (**E**,**F**), in control (circles), FL1 (squares), FL2 (triangles) and DU6P (crosses). Different FAs are represented using different dimensions and tonalities. Data are illustrated as the mean ± SEM (standard error of the mean) and designated significant if *p* < 0.05 (Bonferroni posttest: * *p* < 0.05; ** *p* < 0.01; *** *p* < 0.005, in comparison with control of the same group).

**Figure 11 ijms-23-10245-f011:**
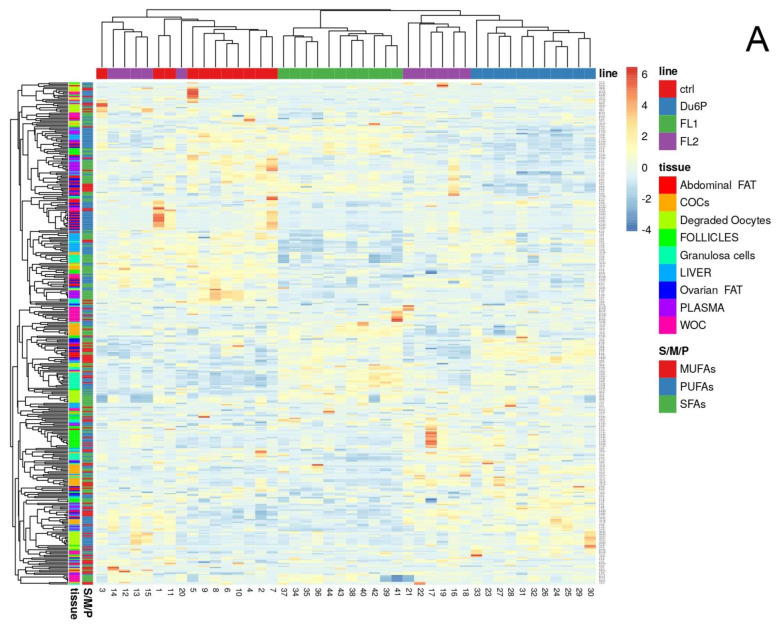
Heatmap (**A**) of all the types of FAs (S/M/P: SFAs, MUFAs and PUFAs, in red, blue and green) in all the analyzed tissues (different tissues in different colors) and principal component analysis (PCA) (**B**) in control (red), DU6P (blue), FL1 (green) and FL2 (purple).

## Data Availability

Original data set is available as Appendix A.

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
