# Peer review of "Lower Plasmatic Levels of Saturated Fatty Acids and a Characteristic Fatty Acid Composition in the Ovary Could Contribute to the High-Fertility Phenotype in Dummerstorf Superfertile Mice"

_ijms, 2022, doi:10.3390/ijms231810245_

Round 1
Reviewer 1 Report
This study investigated the fat content and fatty acid composition of several parts of the female mouse body in two unique mouse lines FL1 and FL2. According to the authors, FLs showed higher body weight and increased body fat content, but at the same time, they managed to decrease the lipid content in the ovarian fat compared to the abdominal fat. In addition, the authors illustrated the differences in fatty acid composition in those tissues, especially a lower level of saturated fatty acids in plasma and a different lipid microenvironment of the ovary. However, the manuscript structure is very confusing. Introduction is too long and missing logical connections. Results are missing the explanation of why experiments were performed. Finally, the discussion does not explain well what causes these differences.
Major Comments
1) Both FL1 and FL2 showed higher reproductive performances, but why the fatty acid composition makes such a large discrepancy? Is it related to the atypical levels of hormones, such as insulin in FL1 and leptin in FL2? The authors did not explain well.
2) The results are very confusing, and it is hard to draw a clear conclusion. In addition, the fatty acid composition of the ovarian microenvironment (Fig. 5B, D and F & Fig. 10) was suggested to be stated together.
3) It would be helpful to include the not shown data as supplemental information.
4) The abstract is not detailed enough to describe the direct results of this study.
5) Figure 5 to Figure 10 were not clear enough.
Minor Comments
1) Results 2.1: the body weight of the mice was measured at 42-60 d, or 42 and 60 d respectively?
2) Paragraph 3 of 2.4: change “Fig.5 (in B, D and E)” to “Fig.5 (in B, D and F)”
3) Line 4 of 2.6: “decreasing SFA PA in FL1 and SA in DU6P” is contrary to the illustration in Fig. 8A.
4) Line 2 of 2.8: change “we have significantly higher levels of SFAs” to “levels of SFAs were significantly higher”
5) P7: the full name of POFs was not marked
6) Are 4.4 and 4.5 the two different methods described in 4.3? If so, it should be marked with 4.3.1 and 4.3.2.
7) Reference 7 volume and page missing. Reference 15 and 19 page missing.
Author Response
Reviewer: 3
This study investigated the fat content and fatty acid composition of several parts of the female mouse body in two unique mouse lines FL1 and FL2. According to the authors, FLs showed higher body weight and increased body fat content, but at the same time, they managed to decrease the lipid content in the ovarian fat compared to the abdominal fat. In addition, the authors illustrated the differences in fatty acid composition in those tissues, especially a lower level of saturated fatty acids in plasma and a different lipid microenvironment of the ovary. However, the manuscript structure is very confusing. Introduction is too long and missing logical connections. Results are missing the explanation of why experiments were performed. Finally, the discussion does not explain well what causes these differences.
- Response: We would like to thank the reviewer for his comment. We partially rewrote the discussion section. However, in our opinion the information included in the introduction section is important to a better understanding of our work. Moreover, we created a graphical abstract for a better understanding of the entire work.
Major Comments
1) Both FL1 and FL2 showed higher reproductive performances, but why the fatty acid composition makes such a large discrepancy? Is it related to the atypical levels of hormones, such as insulin in FL1 and leptin in FL2? The authors did not explain well.
Response: The reviewer is right, indeed the differences in fatty acids analysis in FL1 and FL2 are very different. This discrepancy is correlated to many causes, s those FLs have been selected for around 50 years separately, even following the same criteria. After the selection those fertility lines reached a double ovulation rate, and consequently a double litter size compared to the unselected ctrl line, but they are completely different in many levels, from behavioral, to endocrinal, from physiological to phenotypical. As such, we wanted to analyze the microenvironment of the ovary to figure out if some similarities could have been found at that level. After the analysis, we found that those fertility lines are different at lipid level as well, even showing increased ovulation rate. We should take in account that 100% of the FL1 mice show at least one poly-ovulatory follicle per ovary containing more than one oocyte, which can make a large difference in the selection of the fatty acids from those oocytes. We speculate that they could use a different amount of fat or different types of fatty acids to select which oocyte grows more, or better, or maybe to get enough energy for both of them. Moreover, from the heatmap and APC analysis it is possible to get a general idea concerning the differences between our selection lines. Indeed even comparing the lines only based the lipid analysis, and their fatty acid composition, the correct clustering of individual animals to their particular line is easily detectable.
2) The results are very confusing, and it is hard to draw a clear conclusion. In addition, the fatty acid composition of the ovarian microenvironment (Fig. 5B, D and F & Fig. 10) was suggested to be stated together.
Response: The reviewer is right; we changed the figure statement and partially rewrote the conclusion section. (See section 5).
3) It would be helpful to include the not shown data as supplemental information.
Response: We included an Excel file with all the concentrations as supplementary data.
4) The abstract is not detailed enough to describe the direct results of this study.
Response: The reviewer is right; we added some more details to the abstract (see page 1). Moreover, we created a graphical to make it easier to follow.
5) Figure 5 to Figure 10 were not clear enough.
Response: We created new figures in TIFF format and changed them in the word document as well.
Minor Comments
- Results 2.1: the body weight of the mice was measured at 42-60 d, or 42 and 60 d respectively?
Response: The body weight of the mice was measured at 42, 60 and 75 days (fig. 1.2), after that we measured the weight gain from day 42 to 60 and from day 42 to 75 (fig. 1.2) to get a percentage of growing. (See paragraph 1 of 4.2).
- Paragraph 3 of 2.4: change “Fig.5 (in B, D and E)” to “Fig.5 (in B, D and F)”
Response: The reviewer is right; we changed the sentence. (See paragraph 3 of 4.2)
- Line 4 of 2.6: “decreasing SFA PA in FL1 and SA in DU6P” is contrary to the illustration in Fig. 8A.
Response: The reviewer is right; we changed the sentence. (See line 4 of 2.6).
- Line 2 of 2.8: change “we have significantly higher levels of SFAs” to “levels of SFAs were significantly higher”
Response: We fixed the sentence. (See lines 2 and 3 of 2.8)
- P7: the full name of POFs was not marked
Response: The reviewer is right; we fixed the sentence adding the full name for POFs (poly-ovular follicles). (See line 2 of page 8).
- Are 4.4 and 4.5 the two different methods described in 4.3? If so, it should be marked with 4.3.1 and 4.3.2.
Response: The reviewer is right; we changed the numbering from 4.3. to 4.6.
- Reference 7 volume and page missing. Reference 15 and 19 page missing.
Response:
Calanni-Pileri, M., J. M. Weitzel, M. Langhammer, and M. Michaelis. 2022 b. "Higher quality rather than superior quantity of oocytes determine the amount of fertilizable oocytes in two outbred Dummerstorf high-fertility mouse lines." Reprod Domest Anim. doi: 10.1111/rda.14194. Volume and page are still missing because the article has not yet assigned to an issue.
Czumaj, A., and T. Sledzinski. 2020. "Biological Role of Unsaturated Fatty Acid Desaturases in Health and Disease." Nutrients 12 (2):365-390. doi: 10.3390/nu12020356.
We checked the other references.
CALANNI-PILERI, M., WEITZEL, J. M., LANGHAMMER, M. & MICHAELIS, M. 2022a. Higher quality rather than superior quantity of oocytes determine the amount of fertilizable oocytes in two outbred Dummerstorf high-fertility mouse lines. Reprod Domest Anim.
CALANNI-PILERI, M., WEITZEL, J. M., LANGHAMMER, M., WYTRWAT, E. & MICHAELIS, M. 2022b. Altered insulin, leptin, and ghrelin hormone levels and atypical estrous cycle lengths in two highly-fertile mouse lines. Reprod Domest Anim, 57, 577-586.
LUDWIG, C. L. M., BOHLEBER, S., REBL, A., WIRTH, E. K., VENUTO, M. T., LANGHAMMER, M., SCHWEIZER, U., WEITZEL, J. M. & MICHAELIS, M. 2022. Endocrine and molecular factors of increased female reproductive performance in the Dummerstorf high-fertility mouse line FL1. J Mol Endocrinol, 69, 285-298.
Reviewer 2 Report
The manuscript presents an interesting study about the correlation between the fat weight, abdominal fat content, the fatty acid composition of several parts of the female mouse body and ovulation rate and higher fertility of some high-fertility mouse lines and control. This research could be applied to the field of farm animal biology as well as human reproductive medicine.
Observations
- The writing of the references in the text should be checked and corrected
- Page 9 - 4.1 –“ Pellet ssniff M-Z autoclavable concentrate (Soest, Germany) is composed of N-free extracts (50.1%), starch (34%), crude protein (22%), sugar (5%), crude fat (4.5%), crude fiber (3.9%), and a mineral mixture (3.2%)." - the expression should be checked and changed because as it is written the sum of the percentages of the components exceeds 100
- Fig 2 – it should be explained how abdominal fat weight/body weight and abdominal fat (fat content) – (probably abdominal fat/fat content) were calculated, because in the graph they appear as percentages - for example: (abdominal fat/fat content) x 100
- Fig 3 C – liver (fat content) – without brackets
- Since there are a lot of data resulting from the analyzes for the 4 lines of mice including control, perhaps a table with the representative results should be made to summarize the data obtained
- The conclusions are too general, I think they should be presented in more detail.
Author Response
Reviewer: 1
The manuscript presents an interesting study about the correlation between the fat weight, abdominal fat content, the fatty acid composition of several parts of the female mouse body and ovulation rate and higher fertility of some high-fertility mouse lines and control. This research could be applied to the field of farm animal biology as well as human reproductive medicine.
- Response: We would like to thank the reviewer for his comment.
Observations
- The writing of the references in the text should be checked and corrected
Response: We changed the reference style as suggested.
- Page 9 - 4.1 –“ Pellet ssniff M-Z autoclavable concentrate (Soest, Germany) is composed of N-free extracts (50.1%), starch (34%), crude protein (22%), sugar (5%), crude fat (4.5%), crude fiber (3.9%), and a mineral mixture (3.2%)." - the expression should be checked and changed because as it is written the sum of the percentages of the components exceeds 100
Response: The reviewer is right. We checked the sentence and fixed it as “Pellet ssniff M-Z autoclavable concentrate (catalogue number V1124-300, Soest, Germany) is composed of12% fat, 27% protein and 61% carbohydrates (nitrogen-free-extracts 51.2%)”. However, the N-free extracts are not part of the 100% counting. (See paragraph 4 of 4.1).
- Fig 2 – it should be explained how abdominal fat weight/body weight and abdominal fat (fat content) – (probably abdominal fat/fat content) were calculated, because in the graph they appear as percentages - for example: (abdominal fat/fat content) x 100
Response: The ratio abdominal fat weight/body weight were calculated dividing the weight of the abdominal fat (adipose tissue in the abdomen) by the weight of the entire mouse in the same day it was sacrificed. For example: if the weight of the mouse was 30 gr and the weight of the fat coming from the abdomen was 3 gr, then the ratio is 0.1 (which means that the abdominal fat makes the 10% of the entire body weigh). (See paragraph 2 of 4.2).
“Abdominal fat (fat content)” means “fat content of the abdominal fat” and corresponds to the lipid fraction of that adipose tissue of the abdomen, as it is made by other fractions as well. The same happens with “ovarian fat (fat content)”, “plasma (fat content)” and “liver (fat content)”. They were measured extracting the lipids from the entire tissue (adipose tissue, liver or plasma) and measuring their weight in proportion to 100 mg of tissue or 0.5 ml of plasma. We changed the capture of the figures (see figures 2, 3 and 4 on pages 13-14)
- Fig 3 C – liver (fat content) – without brackets
Response: This correspond to the fat content in 100 mg of liver. We re-wrote the caption of the figures as suggested. (see figures 2, 3 and 4 on pages 13-14)
- Since there are a lot of data resulting from the analyzes for the 4 lines of mice including control, perhaps a table with the representative results should be made to summarize the data obtained
Response: The reviewer is right; so many data were presented in the result part, which is the reason why we used the graphic description instead of the tables, just to make them more clear and readable. We added a supplementary table in order to summarize all data.
6- The conclusions are too general. I think they should be presented in more detail.
Response: We re-wrote the entire conclusion paragraph in order to address this issue. (See section 5, page 11).
Reviewer 3 Report
1. The whole manuscript needs excessive English editing
2. Please, write the Animal care, sample size, and experimental design in a more clear and precise manner and use graphical abstract for the experimental design
3. The estrus synchronization should be rewritten in detail
4. The reference style is not appropriate to the journal guidelines
5. What the authors depending on the choice of oocyte selection
6. How many samples were used
7. What do you mean by Nine samples per animal were used for lipid extraction?
8. The resolution of all figures should be improved
9. The presenting study needs further investigation at the molecular level
Author Response
- The whole manuscript needs excessive English editing
Response: We used the English editing service before the first submission. However, we now switched to the recommended editing service by MDPI, as suggested by the reviewer (changes tracked in two different colors, minor changes correspond to the English editing).
- Please, write the Animal care, sample size, and experimental design in a more clear and precise manner and use graphical abstract for the experimental design.
Response: The reviewer is right. The graphical abstract makes the experimental design easier to follow. We followed the instruction of the reviewer and partially rewrote the paragraph. (See paragraph 3 and 4 of 4.1, and section 4.2).
- The estrus synchronization should be rewritten in detail.
Response: The reviewer is right; we added a sentence to the paragraph 4.2. However, we recently explained the entire concept in detail in our publication, which we cited in the manuscript. See Calanni-Pileri et al. (2022b) for the details.
- The reference style is not appropriate to the journal guidelines.
Response: The reviewer is right. We changed the reference style as suggested.
- What the authors depending on the choice of oocyte selection.
Response: We provided basic description in the present manuscript (See the last 6 lines of 4.2); a more detailed description of the method is provided in Calanni-Pileri et al. (2022a).
- How many samples were used?
Response: We used 11 mice per line (44 in total). From each mouse, we took 9 types of samples (11x4x9=396 samples). (See section 4.3)
- What do you mean by Nine samples per animal were used for lipid extraction?
Response: The reviewer is right, the sentence was not clear. We mean 9 types of samples per animal: plasma, liver, abdominal fat, ovarian fat, small follicles (around 80 μm), COCs, oocytes without cumulus, degraded oocytes and granulosa cells from antral follicles. (See section 4.3)
- The resolution of all figures should be improved.
Response: The reviewer is right; we sent the new pictures in TIFF format. In addition, we changed the figures in the word document as well.
- The presenting study needs further investigation at the molecular level
Response: The reviewer is right; our high fertility mouse lines represent an interesting tool for further molecular investigations, mainly concerning the ovarian microenvironment, as it has already established that their litter size has increased during the generations together with the ovulation rate. We would like to continue the investigations on this innovative and informative mouse line, analyzing other types of organic fractions as well. However, we started from lipids as they can be used as energetic source to get oocytes of higher quality. In addition, some more data can be found in
CALANNI-PILERI, M., WEITZEL, J. M., LANGHAMMER, M. & MICHAELIS, M. 2022a. Higher quality rather than superior quantity of oocytes determine the amount of fertilizable oocytes in two outbred Dummerstorf high-fertility mouse lines. Reprod Domest Anim.
CALANNI-PILERI, M., WEITZEL, J. M., LANGHAMMER, M., WYTRWAT, E. & MICHAELIS, M. 2022b. Altered insulin, leptin, and ghrelin hormone levels and atypical estrous cycle lengths in two highly-fertile mouse lines. Reprod Domest Anim, 57, 577-586.
LUDWIG, C. L. M., BOHLEBER, S., REBL, A., WIRTH, E. K., VENUTO, M. T., LANGHAMMER, M., SCHWEIZER, U., WEITZEL, J. M. & MICHAELIS, M. 2022. Endocrine and molecular factors of increased female reproductive performance in the Dummerstorf high-fertility mouse line FL1. J Mol Endocrinol, 69, 285-298.
Round 2
Reviewer 1 Report
The authors responded to my requests.
Reviewer 3 Report
The author addressed all the comments given by the reviewer adequately.